# Machine-Learning-Based Functional Time Series Forecasting: Application to Age-Specific Mortality Rates

**Ufuk Beyaztas** [1] and **Hanlin Shang** [2,*]

1   Department of Statistics, Marmara University, Istanbul 34722, Turkey; ufuk.beyaztas@marmara.edu.tr
2   Department of Actuarial Studies and Business Analytics, Macquarie University, Sydney, NSW 2109, Australia
*   Correspondence: hanlin.shang@mq.edu.au; Tel.: +61-(2)-9850-4689

**Abstract:** We propose a functional time series method to obtain accurate multi-step-ahead forecasts for age-specific mortality rates. The dynamic functional principal component analysis method is used to decompose the mortality curves into dynamic functional principal components and their associated principal component scores. Machine-learning-based multi-step-ahead forecasting strategies, which automatically learn the underlying structure of the data, are used to obtain the future realization of the principal component scores. The forecasted mortality curves are obtained by combining the dynamic functional principal components and forecasted principal component scores. The point and interval forecast accuracy of the proposed method is evaluated using six age-specific mortality datasets and compared favorably with four existing functional time series methods.

**Keywords:** direct prediction strategy; dynamic functional principal component analysis; long-run covariance; machine learning; recursive prediction strategy

## 1. Introduction

In many developed countries, increases in longevity and an aging population have led to concerns regarding the sustainability of pensions, healthcare, and aged-care systems (Organization for Economic Co-Operation and Development (OECD), 2013) [1]. These concerns have resulted in a surge of interest among government policymakers and planners to accurately model and forecast age-specific mortality rates. In addition, forecasted mortality rates are an important input for determining fixed-term or lifelong annuity prices and are very important to the pension and insurance industries (see, e.g., Shang and Haberman, 2017) [2]. In demography, many statistical methods have been proposed for forecasting age-specific mortality rates (see, Booth and Tickle, 2008, Shang et al. 2011, for reviews) [3,4]. Of these, the most famous benchmark model is the Lee and Carter's (1992) [5] model. This model uses a principal component method to extract a single time-varying index from the data to model the age-specific mortality rates. Then, the forecasts are obtained via a random walk with drift. The LC method has the characteristics of simplicity and robustness when the age-specific $\log_e$ mortality rates have linear trends (see, e.g., Booth et al., 2006) [6]. On the other hand, it uses only one principal component and its associated scores to capture the mortality patterns in the data. The use of only one principal component to decompose mortality rates may not capture the majority of variability in data. Thus, the optimal forecasts may not be obtained by the LC method. To overcome this problem, several methods, which are the extensions of the LC method, have been proposed (see, e.g., Booth et al., 2002, Renshaw and Haberman, 2003, Cairns et al., 2006, Renshaw and Haberman, 2006, Cairns et al., 2009, Plat, 2009, Hatzopoulos and Haberman, 2009, Hunt and Blake, 2014, Wiśniowski et al., 2015) [7–15]. In addition to the classical time series methods, several machine learning and deep learning methods have been extended to the mortality forecasting (see, e.g., Deprez et al., 2017, Richman and Wüthrich 2021, Perla et al., 2021) [16–18].

In addition to the aforementioned LC-based models, several functional time series (FTS) methods have been proposed to obtain improved forecasts for the age-specific mortality rates (see, e.g., Hyndman and Ullah, 2007, Hyndman et al., 20013, Gao and Shang, 2017, Shang and Haberman, 2018, Shang, 2019, Shang, 2020, Shang and Yang, 2021 [19–25] and references therein). In the FTS method, contrary to the LC-based methods, the age-specific mortality curves, where age is treated as a continuum, are analyzed. In more detail, the FTS method decomposes the smooth mortality curves into a set of functional principal components and associated scores via a functional principal component analysis (FPCA) method. Then, the principal component scores are modeled using a classical time series method to obtain their future realizations. Finally, the forecasted mortality curves are obtained by combining the future realizations of the principal component scores and the functional principal components. Obtaining accurate forecasts for the future realizations of the principal component scores is crucial for the FTS methods to have improved forecasting results. All the existing FTS forecasting methods for mortality curves use classical linear time series methods to obtain future realizations of the principal component scores. However, classical time series methods are data-dependent and require several assumptions for the distribution of the error process. In addition, such classical time series methods assume that the principal component scores belong to a true data-generation process. Thus, different principal component scores for different mortality datasets may require different estimation strategies for the parameters of the assumed model. Moreover, in most existing FTS methods, the majority of attention has been paid to a one-step-ahead forecast of the mortality curves. On the other hand, the one-step-ahead forecast may not be informative because the decision-makers and policymakers may need long-term forecasts of mortality rates to manage risks properly. Therefore, in contrast to the one-step-ahead forecast, $h$-step-ahead ($h > 1$) mortality forecasts may be more useful for policymakers to make long-term plans easily. In this paper, we propose a machine-learning-based FTS method, which automatically uses learning algorithms to learn the underlying structure of the principal component scores to obtain $h$-step-ahead forecasts of mortality curves.

In our proposed method, contrary to most of the FTS methods that use the static FPCA, we consider a dynamic FPCA (DFPCA) method to decompose the mortality curves into the principal components and the corresponding scores. In DFPCA, the principal components are extracted based on an estimated long-run covariance, including the variance function as a component. It also measures temporal cross-covariance at different positive and negative lags. Thus, compared to the static FPCA, the DFPCA produces more consistent principal components (see, e.g., Shang, 2019, Shang, 2020 [23,24]). To obtain the future realizations of the principal component scores, we consider three machine-learning-based multi-step-ahead time series forecasting strategies: (1) recursive strategy, which iteratively uses the one-step-ahead forecast procedure for all the forecast horizons; (2) direct strategy, which considers different models for a forecast horizon; and (3) DirRec strategy, which combines the recursive and direct strategies to obtain $h$-step-ahead forecasts effectively. Consult Sorjamaa et al., 2007, Taieb et al., 2012, Taieb and Hyndman, 2014 [26–28] for more information about the multi-step-ahead time series forecasting strategies. In addition to the point forecast, we adopt the proposed method into the bootstrap method of Hyndman and Ullah 2007 [19] to construct pointwise prediction intervals for the mortality curves.

The remaining part of this paper is organized as follows. Section 2 summarizes the FTS and DFPCA methods and introduces the multi-step-ahead forecasting strategies. The datasets used to evaluate the forecast accuracy of the proposed methods are presented in Section 3. In Section 4, we revisit the expanding-window approach and forecast accuracy measures to evaluate the accuracy of the proposed methods. The results are presented in Section 5. Section 6 concludes the paper, along with some ideas on how the method can be further extended.

## 2. Functional Time Series Forecasting Method

Let $\boldsymbol{\mathcal{X}}(u) = \{\mathcal{X}_1(u), \ldots, \mathcal{X}_n(u)\}$ denote a functional time series of the log mortality rate of age $u$, where $u$ denotes the age continuum. It is assumed that the functions $\mathcal{X}_i(u)$, for $i = 1, \ldots, n$, are the elements of $\mathcal{L}_2(\mathcal{I})$ Hilbert space defined on the closed and bounded interval $\mathcal{I}$, i.e., $u \in \mathcal{I}$. The direct modeling and forecasting of functional time series are difficult tasks because the elements of such time series belong to an infinite-dimensional Hilbert space. A common practical solution to overcome this problem is the projection of the infinite-dimensional time series into a finite-dimensional space of basis functions. The commonly used approach for this purpose is dimension reduction method.

Let $\mu(u) = \mathrm{E}[\mathcal{X}(u)]$ denote the mean function and $C(u,s) = \mathrm{Cov}[\mathcal{X}(u), \mathcal{X}(s)]$ represents the covariance function of $\mathcal{X}(u)$ satisfying $\int_{\mathcal{I}} \int_{\mathcal{I}} C^2(u,s) du ds < \infty$. By Mercer's Theorem, the covariance function has the following representation

$$C(u,s) = \sum_{k \geq 1} \lambda_k \phi_k(u) \phi_k(s), \quad u, s \in \mathcal{I},$$

where $\phi_k(u)$ and $\phi_k(s)$, for $k = 1, 2, \ldots$, denote the orthonormal principal components corresponding to the non-negative eigenvalues $\lambda_k$. Then, by the Karhunen–Loève expansion, a stochastic process $\mathcal{X}(u)$ can be expressed as

$$\mathcal{X}(u) = \mu(u) + \sum_{k=1}^{\infty} \beta_k \phi_k(u), \tag{1}$$

where $\beta_k$s are the principal component scores obtained by the projection of $\mathcal{X}(u) - \mu(u)$ in the direction of the $k^{\text{th}}$ eigenfunction $\phi_k$, i.e., $\beta_k = \langle \mathcal{X}(u) - \mu(u), \phi_k(u) \rangle$.

Dimension reduction can be achieved by truncating the infinite series expansion in Equation (1) to the first $K$ functional principal components. The main feature in the original functional time series can be captured by $K$-dimensional vector $(\beta_1, \beta_2, \ldots, \beta_K)$, leading to an approximation as

$$\mathcal{X}(u) = \mu(u) + \sum_{k=1}^{K} \beta_k \phi_k(u) + e(u), \tag{2}$$

where $e(u)$ denotes the error term containing those principal components and their associated scores, excluded from the first $K$ truncation. The forecasting performance of the functional time series methods is significantly affected by choice of $K$. In statistics, several approaches, such as cross-validation (Ramsay and Silverman, 2006) [29], Akaike information criterion (Akaike, 1974) [30], and explained variance (Chiou, 2012) [31] can be used to determine the optimum value of $K$. In our analyses, we consider the cross-validation approach to determine the optimum value of $K$.

### 2.1. Dynamic Functional Principal Component Analysis

In Equations (1) and (2), the principal components are obtained by maximizing the variance information from the data. They enjoy optimality features (see Shang, 2014 for independent and identically distributed functional data). For FTS with moderate-to-strong temporal dependence, the variance information may not be an adequate criterion as it does not incorporate autocovariance at different lags in an FTS. As an alternative, we apply a DFPCA constructed from an eigendecomposition of an estimated long-run covariance function. The long-run covariance function includes the variance and autocovariance at lags greater than zero.

To estimate the long-run covariance function from an FTS, we consider a kernel sandwich estimator

$$\widehat{C}_{h,q}(s,u) = \sum_{\ell=-\infty}^{\infty} W_q\left(\frac{\ell}{h}\right) \widehat{\gamma}_\ell(s,u), \tag{3}$$

where $\ell$ denotes the lag operator, $\widehat{\gamma}_\ell(s, u)$ denotes an estimator of the empirical autocovariance function at lag $\ell$

$$\widehat{\gamma}_\ell(s, u) = \begin{cases} \frac{1}{n} \sum_{i=1}^{n-\ell} [\mathcal{X}_i(u) - \overline{\mathcal{X}}(u)][\mathcal{X}_{i+\ell}(s) - \overline{\mathcal{X}}(s)] & \text{if } \ell \geq 0, \\ \frac{1}{n} \sum_{i=1-\ell}^{n} [\mathcal{X}_i(u) - \overline{\mathcal{X}}(u)][\mathcal{X}_{i+\ell}(s) - \overline{\mathcal{X}}(s)] & \text{if } \ell < 0, \end{cases}$$

and $W_q(\frac{\ell}{h})$ represents a weight function. The weight function depends on the order of kernel function and bandwidth parameter $h$. The bandwidth parameter plays an important role in estimating the long-run covariance function. In this study, we consider the plug-in algorithm proposed by Rice and Shang (2017) [32] to select the optimal value of the bandwidth.

With the estimated long-run covariance function $\widehat{C}_{h,q}(s, u)$, the eigen-decomposition is used to extract the principal components and their associated scores. Let $\{\widehat{\lambda}_i : i = 1, 2, \ldots\}$, for $\widehat{\lambda}_i > \widehat{\lambda}_{i+1}$, denote the sample eigenvalues of $\widehat{C}_{h,q}(s, u)$ and $\widehat{\boldsymbol{\Phi}}(u) = [\widehat{\phi}_1(u), \widehat{\phi}_2(u), \ldots,]$ denotes the corresponding orthogonal sample eigenfunctions. By the Karhunen–Loève expansion, we have the following representation for the stochastic process $\mathcal{X}(u)$

$$\mathcal{X}(u) = \widehat{\mu}(u) + \sum_{k=1}^{\infty} \widehat{\beta}_k \widehat{\phi}_k(u),$$

where $\widehat{\mu}(u) = \frac{1}{n} \sum_{i=1}^{n} \mathcal{X}_i(u)$ and $\widehat{\beta}_k$ is the $k^{\text{th}}$ estimated dynamic principal component score. After the DFPCA decomposition of the FTS (with first $K$ dynamic principal components), the conditional expectation results in the $h$-step-ahead point forecast of $\mathcal{X}_{n+h}(u)$ as follows

$$\widehat{\mathcal{X}}_{n+h|n}(u) = \mathrm{E}[\mathcal{X}_{n+h}(u)|\widehat{\mu}(u), \widehat{\boldsymbol{\Phi}}(u), \boldsymbol{\mathcal{X}}(u)]$$

$$= \widehat{\mu}(u) + \sum_{k=1}^{K} \widehat{\beta}_{n+h|n,k} \widehat{\phi}_k(u),$$

where $\widehat{\beta}_{n+h|n,k}$ denotes the $h$-step-ahead point forecast of $\beta_{n+h,k}$.

### 2.2. Multi-Step-Ahead Time Series Forecasting Strategies

To obtain the forecasts of principal component scores $[\widehat{\beta}_{n+h|n,1}, \widehat{\beta}_{n+h|n,2}, \ldots, \widehat{\beta}_{n+h|n,K}]$, we consider three multi-step-ahead time series forecasting strategies, i.e., recursive, direct, and DirRec. For each strategy, we assume that the time series of principal component scores, $\{\beta_{k,1}, \ldots, \beta_{k,n}\}$ for $k = 1, 2, \ldots, K$, follows an autoregressive process with a function $f$, a lag order $d$, and an error process $\epsilon_n$ with mean zero and variance $\sigma^2$

$$\beta_{k,n} = f(\boldsymbol{\eta}_{k,n-1}) + \epsilon_{k,n}, \tag{4}$$

where $\boldsymbol{\eta}_{k,n-1} = [\beta_{k,n-1}, \ldots, \beta_{k,n-d}]^{\top}$ and $^{\top}$ denotes matrix transpose. To measure the forecast errors, we consider the mean squared error (MSE). Let $g(\boldsymbol{\eta}_{k,n}; \widehat{\theta}; h)$ denote the $h$-step-ahead forecast obtained from $\boldsymbol{\eta}_{k,n}$ with the estimated parameter vector $\widehat{\theta}$. Then, the MSE is given by

$$\mathrm{MSE}_h(\boldsymbol{\eta}_{k,n}) = \mathrm{E}_{\epsilon, \beta_k} \left\{ \left[ \beta_{k,n+h} - g(\boldsymbol{\eta}_{k,n}; \widehat{\theta}; h) \right]^2 | \boldsymbol{\eta}_{k,n} \right\}. \tag{5}$$

From Equation (5), the optimal $h$-step-ahead forecast, i.e., the forecast having the minimum MSE at forecast horizon $h$, is given by $\mathrm{E}(\beta_{k,n+h}|\boldsymbol{\eta}_{k,n})$.

The *h*-step-ahead forecasts are obtained in recursive strategy by applying a one-step-ahead forecast procedure at each forecast horizon. This procedure is repeated until all the forecasts are obtained. This method is of the form as in Equation (4) and aims to minimize the one-step-ahead forecast variance. In other words, the recursive method estimates the following model

$$\beta_{k,n} = m(\vartheta_{k,n-1}; \kappa) + e_{k,n},$$

where $\vartheta_{k,n-1} = [\beta_{k,n-1}, \ldots, \beta_{k,n-s}]^{\top}$, $s$ is the estimate of true lag parameter $d$, $\kappa = [\rho, \tau]$ is the vector of hyperparameter $\rho$ and model parameter $\tau$, and

$$e_{k,n} = f(\eta_{k,n-1}) - m(\vartheta_{k,n-1}; \kappa) + \epsilon_{k,n}$$

denotes the forecast error. In this method, the estimates of $d$ and $\rho$ are obtained by minimizing the one-step-ahead forecast variance

$$(s, \widehat{\rho}) = \arg\min_{s,\rho} \sum_{(\vartheta_{k,n-1}, \beta_{k,n}) \in D} [\beta_{k,n} - m(\vartheta_{k,n-1}; \widehat{\rho}, \widehat{\tau})]^2. \tag{6}$$

In Equation (6), $\widehat{\tau}$ denotes an estimate of model parameter $\tau$ using the validation set $D$. Based on the one-step-ahead forecast given above, the *h*-step-ahead forecasts are obtained as

$$m^{(h)}(\vartheta; \kappa) = \begin{cases} m(\vartheta; \widehat{\kappa}), & h = 1, \\ m[m^{(h-1)}(\vartheta; \widehat{\kappa}), \ldots, m^{(1)}(\vartheta; \widehat{\kappa}), \beta_{k,n}, \ldots, \beta_{k,n-s+h}, \widehat{\kappa}] & 1 < h \leq s, \\ m[m^{(h-1)}(\vartheta; \widehat{\kappa}), \ldots, m^{(h-s)}(\vartheta; \widehat{\kappa}); \widehat{\kappa}] & h > s. \end{cases}$$

Note that in recursive strategy, different sets of parameter estimates are used at each forecast horizon $h$

$$(s_h, \widehat{\rho}) = \arg\min_{s,\rho} \sum_{(\vartheta_{k,n-h}, \beta_{k,n}) \in D_h} \left[ \beta_{k,n} - m^{(h)}(\vartheta_{k,n-h}; \rho, \widehat{\tau}_h) \right]^2.$$

In direct strategy, different forecasting methods are fitted for each forecast horizon. More precisely, for each forecast horizon, the direct strategy aims to fit a model of the form

$$\beta_{k,n} = m_h(r_{k,n-h}; \gamma_{k,h}) + e_{k,n,h},$$

where $r_{k,n-h} = [\beta_{k,n-h}, \ldots, \beta_{k,n-h-s_h}]^{\top}$, $s_h$ is the estimate of the true lag parameter $d$ at forecast horizon $h$, $\gamma_h = (\rho_h, \tau_h)$ is the vector of hyperparameter $\rho_h$ and model parameter $\tau_h$ at horizon $h$, and $e_{k,n,h}$ is the forecast error obtained from model $m_h$ at forecast horizon $h$. In this method, the estimates of the lag parameter and hyperparameter for model $m_h$ are obtained by minimizing the forecast error as

$$(s_h, \widehat{\rho}_h) = \arg\min_{s,\rho} \sum_{(r_{k,n-h}, \beta_{k,n}) \in D_h} [\beta_{k,n} - m_h(r_{k,n-h}; \rho, \widehat{\tau}_h)]^2, \tag{7}$$

where $\widehat{\tau}_h$ denotes the estimate of model parameter using the validation set $D_h$ at forecast horizon $h$.

When comparing these two strategies, the recursive strategy tries to match the forecasting and assumed models as much as possible. On the other hand, the direct strategy does not. In addition, contrary to the recursive strategy, which minimizes one-step-ahead forecast errors, the direct strategy considers minimizing *h*-step-ahead forecast errors. Moreover, the direct strategy requires more computing time than the recursive strategy, since the former one estimates $h$ models rather than the one model that the recursive strategy estimates.

The DirRec strategy combines the recursive and direct strategies to obtain accurate forecasts. In this method, as in the direct strategy, different forecasting models are used for each forecast horizon $h$, but the forecasts obtained from the forecast horizon $h - 1$ are included in the model constructed at horizon $h$ as input as in recursive strategy. In this strategy, the parameters are estimated as

$$(s_h, \widehat{\rho}_h) = \arg\min_{s,\rho} \sum_n \left\{ \beta_{k,n} - [m_h(\widehat{m}_{h-1}, \dots, \widehat{m}_1, r_{k,n-h}; \rho, \widehat{\tau}_h)] \right\}^2,$$

and the $h$-step-ahead forecasts are obtained as

$$m^{(h)}(\vartheta; \kappa) = \begin{cases} m(\vartheta; \widehat{\kappa}), & h = 1 \\ m[m^{(h-1)}(\vartheta; \widehat{\kappa}), \dots, m^{(1)}(\vartheta; \widehat{\kappa}), \beta_{k,n}, \dots, \beta_{k,n-s+1}, \widehat{\kappa}] & h > 1 \end{cases}$$

Throughout this study, the standard neural network is used for the recursive and direct strategies as the learning algorithm

$$m(\eta) = \tau_0 + \sum_{j=1}^{N} \tau_j g(w_j^\top \eta), \tag{8}$$

where $\eta$ denotes the vector of inputs, $w_j$ is the weight vector for the $j$th hidden node, $\{\tau_0, \tau_1, \dots, \tau_n\}$ are the weights for the output node, and $N$ is the number of hidden nodes. Here, $g(\cdot)$ denotes the output for the hidden note and is obtained via the logistic function, i.e., $g(v) = \frac{1}{1+e^{-v}}$. In Equation (8), the number of hidden nodes $N$ controls the complexity of the model, and the corresponding weights are estimated using optimization techniques, such as backpropagation (see, e.g., Taieb et al., 2012 [27]). The weights are initially chosen close to zero and updated by the backpropagation to minimize the prediction errors. In this method, the results depend on the initial value; thus, the neural network is trained for different initial values. The outputs are obtained by taking the average of different models' outputs. In this study, the neural networks are performed using the R package "nnet" (Venables and Ripley, 2002 [33]). For the DirRec strategy, on the other hand, the linear regression, which is a parametric model, is used as a learning algorithm

$$m(\eta) = \sum_{j=1}^{p} \tau_j \eta_j.$$

Here, the parameter estimates are obtained using the ordinary least squares method.

### 2.3. Construction of Prediction Interval

We consider the nonparametric bootstrap method proposed by Hyndman and Shang (2009) [34] to obtain pointwise prediction intervals for the mortality rates. In this method, two sources of errors are taken into account: (1) the errors in estimating the regression coefficient estimates; and (2) the errors in the model residuals. Using the multi-step-ahead forecasting strategies, we can obtain multi-step-ahead forecasts of the scores, $\{\widehat{\beta}_{k,1}, \dots, \widehat{\beta}_{k,n}\}$ and their associated forecast errors

$$\xi_{k,i,h} = \widehat{\beta}_{k,i} - \widehat{\beta}_{k,i|i-h}, \quad i = h+1, \dots, n.$$

These forecast errors allow us to construct multi-step-ahead bootstrap samples of principal component scores

$$\widehat{\beta}_{k,n+h|n}^b = \widehat{\beta}_{k,n+h|n} + \widehat{\xi}_{*,k,h}^b, \quad b = 1, \dots, B,$$

where $\widehat{\xi}_{*,k,h}^b$ is a random drawn from $\widehat{\xi}_{*,k,h}$ and $B$ denotes the number of bootstrap replications. The residuals in the functional principal component regression can be sampled with

replacement to form the bootstrap samples $\widehat{e}^{b}_{n+h}(u)$. Adding these two sources of errors, we have

$$\widehat{\mathcal{X}}^{b}_{n+h|n}(u) = \widehat{\mu}(u) + \sum_{k=1}^{K} \widehat{\beta}^{b}_{k,n+h|n} \widehat{\phi}_k(u) + \widehat{e}^{b}_{n+h}(u).$$

The $100(1-\alpha)\%$ prediction intervals can be constructed by taking $\alpha/2$ and $(1-\alpha/2)$ empirical quantiles of $\{\widehat{\mathcal{X}}^{1}_{n+h}(u), \dots, \widehat{\mathcal{X}}^{B}_{n+h}(u)\}$.

## 3. Age-Specific Mortality Data Sets

We study the age-specific mortality rates of three countries; Australia (from 1921 to 2018), Canada (from 1921 to 2019), and the United Kingdom (UK) (from 1922 to 2018), obtained from the Human Mortality Database (https://www.mortality.org/ (accessed on 31 January 2022)). The Human Mortality Database includes age-specific mortality rates for 41 countries. These three countries (i.e., Australia, Canada, and the UK) are selected because they have high data quality. In the datasets, the observations are yearly mortality curves from ages 0 to 110+ (110+ denotes ages at and beyond 110). Here, age is treated as the continuum in the mortality curves. For each dataset, we only consider the data from 1950 to 2018 to avoid possible abnormal death rates before 1950 due to the two world wars and the Spanish flu pandemic. There are some years where missing values occur for ages between 96 and 100. In addition, erratic death rates can be observed at and beyond age 95. Thus, we only consider ages from 0 to 95+, where the last age group includes those at and beyond 95. For each sex in a given year, the observed log mortality curves are smoothed via the penalized regression splines with a monotonically increasing constraint after the age of 65 (see, e.g., Hyndman and Ullah, 2007, Hyndman and Shang, 2010 [19,35]). In more detail, we assume that each observed log mortality curve $\mathcal{X}_i(u)$, for $i = 1, \dots, n$, is characterized by a smooth function $\mathcal{F}_i(u)$ and a random error term with mean zero and unit variance $\varepsilon_i$ as

$$\mathcal{X}_i(u_j) = \mathcal{F}_i(u_j) + \sigma_i(u_j)\varepsilon_{i,j}, \quad j = 1, \dots, J,$$

where $j$ denotes the number of observed ages and $\varepsilon_{i,j}$ is a component that allows us to model heteroskedasticity, which can be estimated by

$$\widehat{\sigma}_i(u_j) = \frac{1}{e^{\mathcal{X}_i(u_j)} A_i(u_j)},$$

where $A_i(u_j)$ denotes the exposure-at-risk. With the penalized regression spline with monotonic constraint, the smoothed log mortality curves are obtained as

$$\mathcal{F}_i(u_j) = \operatorname*{argmin}_{\theta_i(u_j)} \sum_{j=1}^{J} w_i(u_j)|\mathcal{X}_i(u_j) - \theta_i(u_j)| + \delta \sum_{j=1}^{J-1} |\theta'_i(u_{j+1}) - \theta'_i(u_j)|,$$

where $\delta$ is the smoothing parameter, $\theta$ is the smooth function approximated from the $B$-splines, and $\theta'$ denotes the first derivative of $\theta$.

The rainbow plots of the smoothed mortality and log mortality rates are present in Figure 1, where the red-colored curves denote the mortality rates for distant-past years and the mortality rates for more recent years are given by violet. From the log mortality plots, for both females and males, it is evident that the mortality rates decrease sharply in infant ages, climb the 20 s, and then linearly increase with age on the log scale.

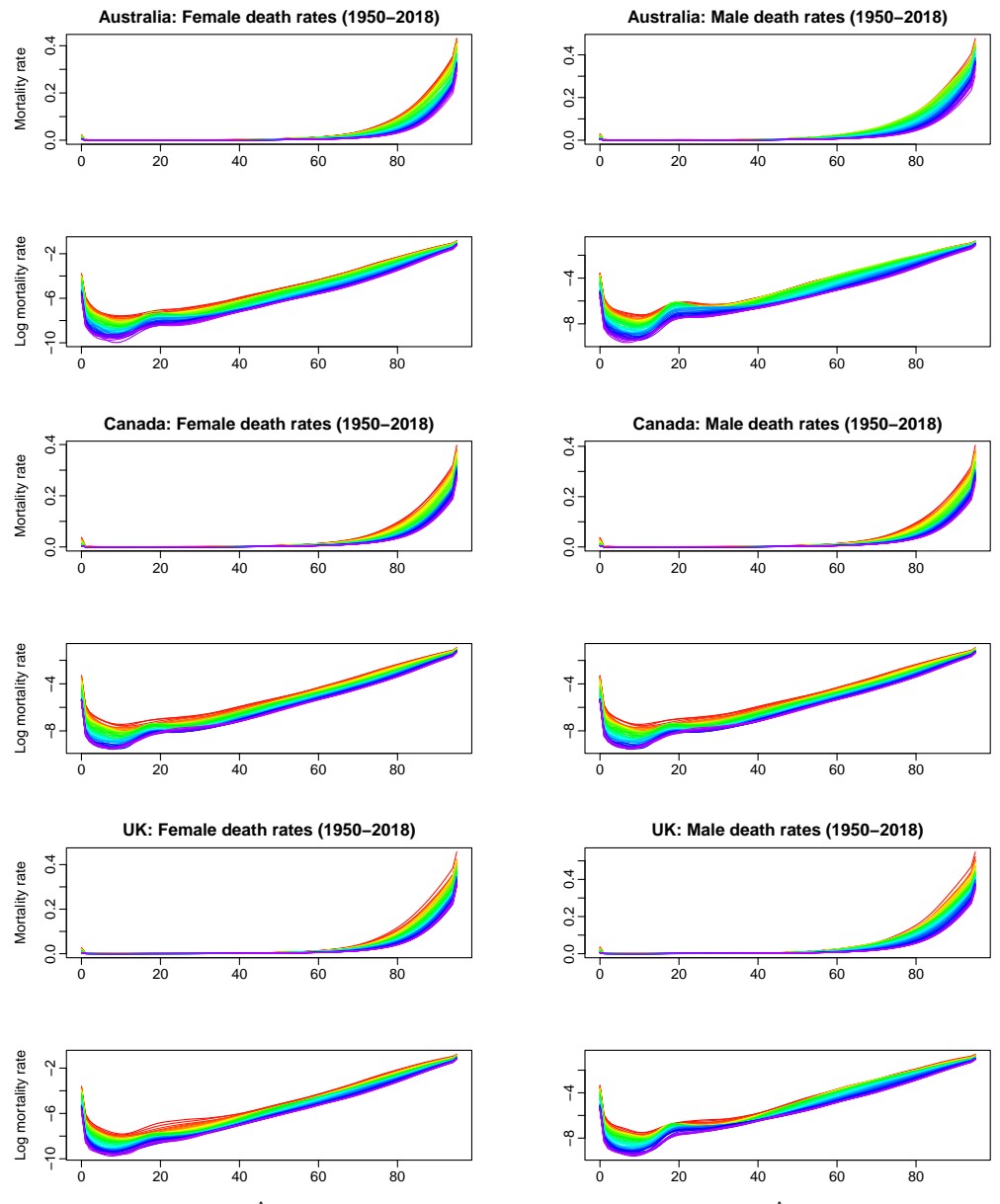

**Figure 1.** Functional time series plots of smoothed mortality and log mortality rates for Australia (rows 1–2), Canada (rows 3–4), and the UK (rows 5–6): female (**left panels**) and male (**right panels**).

## 4. Forecast Evaluation

### 4.1. Expanding-Window Approach

We consider an expanding-window approach to evaluate the forecast accuracy of the proposed methods. We split the entire data into two parts for each dataset: a training sample comprising years from 1950 to 1998 (48 years in total) and a test sample consisting of years from 1999 to 2018 (20 years in total). Using the entire training sample, we obtain $h = 1, \ldots, 20$-step-ahead forecasts of log mortality rates in 1999–2018. Then, we obtain $h = 1, \ldots, 19$-step-ahead forecasts of log mortality rates in 2000–2019 by increasing the training sample one year. We keep iterating until the training sample covers the entire data sample. This procedure produces 20 one-step-ahead, 19 two-step-ahead, ..., and one 20-step-ahead forecast from 1999 to 2018. To evaluate the forecasting accuracies of the methods, we compare the obtained forecasts with the holdout samples.

### 4.2. Measures of Forecast Accuracy

We compute the point forecast accuracy by mean squared forecast error (MSFE) and mean absolute forecast error (MAFE) for each forecast horizon, which measures the squared and absolute differences between forecasts and holdout samples in the testing data, respectively. The MSFE and MAFE can be expressed as

$$\text{MSFE}_i = \frac{1}{J} \sum_{j=1}^{J} [\mathcal{X}_i(u_j) - \widehat{\mathcal{X}}_i(u_j)]^2,$$

$$\overline{\text{MSFE}} = \frac{1}{n} \sum_{i=1}^{N_{\text{test}}} \text{MSFE}_i,$$

$$\text{MAFE}_i = \frac{1}{J} \sum_{j=1}^{J} |\mathcal{X}_i(u_j) - \widehat{\mathcal{X}}_i(u_j)|,$$

$$\overline{\text{MAFE}} = \frac{1}{n} \sum_{i=1}^{N_{\text{test}}} \text{MAFE}_i,$$

where $J$ denotes the number of discrete ages in a log mortality curve, and $N_{\text{test}}$ is the number of observations in the forecast horizon.

To measure the pointwise interval forecast accuracy, we consider the interval score of [36]. The interval scores can be expressed as

$$S_\alpha\left[\widehat{\mathcal{X}}^{\text{lb}}(u_j), \widehat{\mathcal{X}}^{\text{ub}}(u_j); \mathcal{X}(u_j)\right] = \left[\widehat{\mathcal{X}}^{\text{ub}}(u_j) - \widehat{\mathcal{X}}^{\text{lb}}(u_j)\right] + \frac{2}{\alpha}\left[\widehat{\mathcal{X}}^{\text{lb}}(u_j) - \mathcal{X}(u_j)\right]\mathbb{1}\left\{\mathcal{X}(u_j) < \widehat{\mathcal{X}}^{\text{lb}}(u_j)\right\}$$
$$+ \frac{2}{\alpha}\left[\mathcal{X}(u_j) - \widehat{\mathcal{X}}^{\text{ub}}(u_j)\right]\mathbb{1}\left\{\mathcal{X}(u_j) > \widehat{\mathcal{X}}^{\text{lb}}(u_j)\right\},$$

where $[\widehat{\mathcal{X}}^{\text{lb}}(u_j), \widehat{\mathcal{X}}^{\text{ub}}(u_j)]$ denote the lower and upper bounds of a prediction interval, $\mathbb{1}\{\cdot\}$ denotes the binary indicator function, and $\alpha$ represents a level of significance, customarily $\alpha = 0.95$. Averaging over all observations in the forecasting period, we use the mean interval scores $\overline{S}_{\alpha,i}$ to evaluate and compare interval forecast accuracy.

## 5. Mortality Data Analyses

This section presents the forecast accuracy of the proposed machine learning-based FTS methods. Compared with the recursive and DirRect strategies, the direct strategy has the worst forecasting performance, requiring more computing time. Thus, we only present the results for the recursive and DirRect strategies. We compare the forecast accuracy of the proposed methods with four commonly used traditional time series-based FTS methods; exponential smoothing (ETS), random walk (RW), and random walk with drift (RWD) using the R package `ftsa` Hyndman and Shang (2021) [37] and autoregressive integrated moving average (ARIMA) using the R package `demography` Hyndman (2019) [38]. RW fails to produce good forecast accuracy results compared with other methods (its results are even worse than those of direct strategy); thus, its results are discarded from the paper. The R code for the proposed methods can be obtained from the authors upon request.

The point forecast accuracy results computed for each method and all six datasets are presented in Figures 2 and 3. This figure shows that the machine learning-based methods produce similar forecasts (i.e., similar MSFE and MAFE values) to that of the ETS, RWD, and ARIMA-based methods for short and moderate-term forecasts horizons. On the other hand, the use of both multi-step-ahead forecasting strategies results in better forecasts than the classical time series methods for long-term forecasts horizons. Among others, the ETS-based FTS method produces the worst results, i.e., it produces higher MSFE and MAFE values, especially for the long-term forecast horizons. While the RWD- and ARIMA-

based FTS methods produces better MSFE and MAFE results than the ETS-based method, machine learning-based FTS methods generally produce improved forecasts than the RWD- and ARIMA-based method. When the proposed methods are compared, while it seems that the superiorities of the recursive and DirRec strategies over each other vary from data to data, the recursive-based method generally produces improved MSFE and MAFE results than those of the DirRec-based method. In a nutshell, the results reported in Figure 2 indicate that all the methods generally produce similar forecasts up to a five-eight-step-ahead forecast horizon. On the other hand, the proposed machine-learning-based FTS methods produce better forecasts than the classical FTS methods.

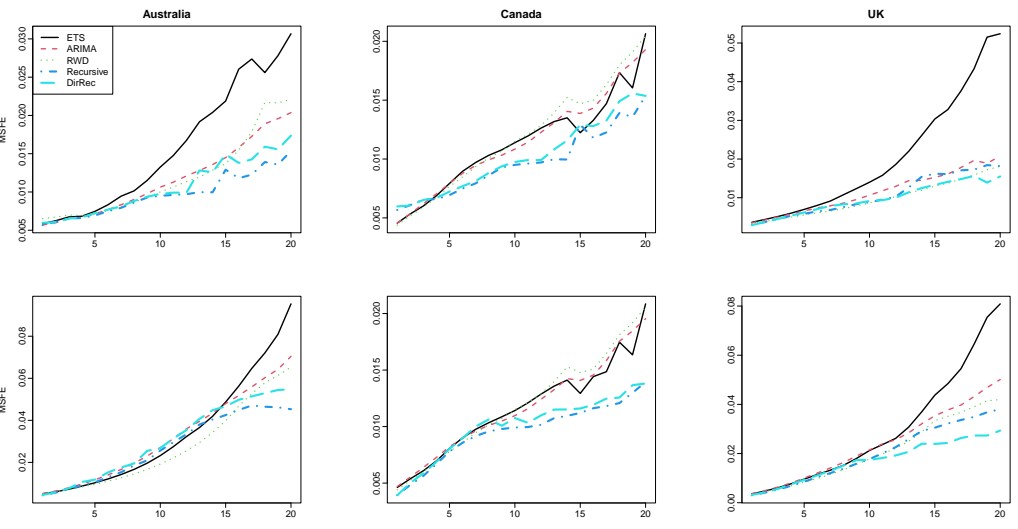

**Figure 2.** Computed $\overline{\text{MSFE}}$ values of log mortality rates for Australia (**first column**), Canada (**second column**), and UK (**third column**). The results for the female group are given in the first row while the results for the male group are given in the second row.

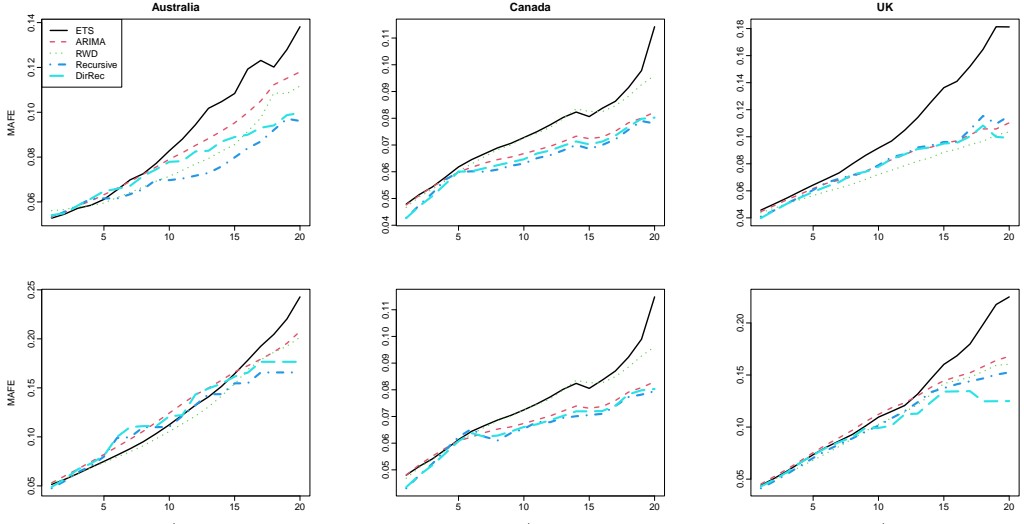

**Figure 3.** Computed $\overline{\text{MAFE}}$ values of log mortality rates for Australia (**first column**), Canada (**second column**), and UK (**third column**). The results for the female group are given in the first row while the results for the male group are given in the second row.

To support our findings, we present 20-step-ahead forecast errors functions and the histograms of pointwise 20-step-ahead forecast errors obtained from the Australian female log mortality dataset in Figures 4 and 5, respectively. From these figures, the error functions obtained by the recursive strategy are closer to zero than those of other methods. In other

words, the proposed recursive-based FTS method comparably produces smaller forecast errors than other methods. The proposed DirRec and the existing RWD- and ARIMA-based methods produce similar error functions, resulting in the second-best methods. The ETS-based FTS method produces the worst (i.e., furthest from zero). In addition, from Figure 4, it is evident that the error functions obtained from the proposed recursive-based FTS method follow a more similar structure compared with those of other methods. In addition, compared with other methods, the error functions of the recursive-based FTS method for the age range 0–20 show less variation. These results indicate that the proposed recursive-based FTS method produces a more consistent forecast than the other methods. From the histograms of the pointwise forecast errors (see Figure 5), the distribution of the pointwise error terms obtained from the proposed methods shows a more similar structure to the normal distribution with mean zero than the distribution of the pointwise errors obtained by the classical FTS methods. Given that the error process in an FTS is assumed to follow a normal distribution with mean zero (see, e.g., Section 3), the results presented in Figures 4 and 5 indicate that the proposed methods not only produce improved forecast (i.e., smaller forecast errors) than the traditional FTS methods, but they also fulfill the underlying model assumption.

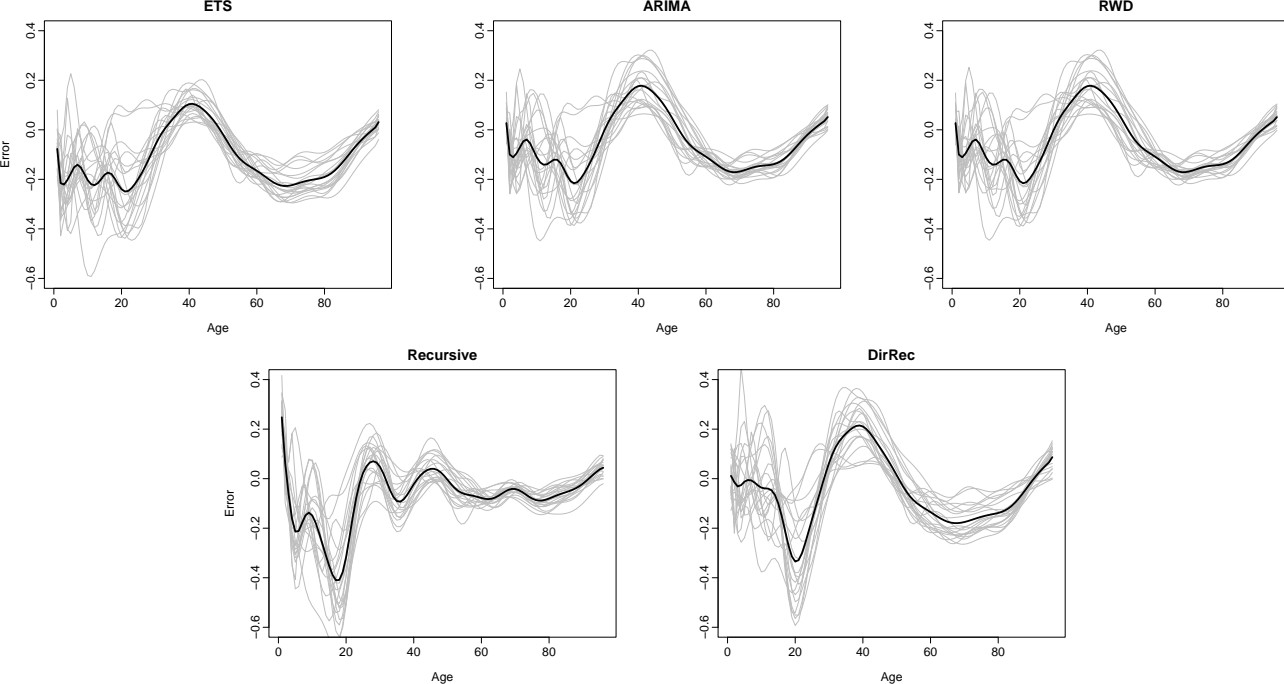

**Figure 4.** Plots of the 20-step-ahead forecast errors functions (gray lines denote the error functions and black lines denote the mean of the forecast error functions) were obtained by all the methods for the log mortality dataset of Australia (female).

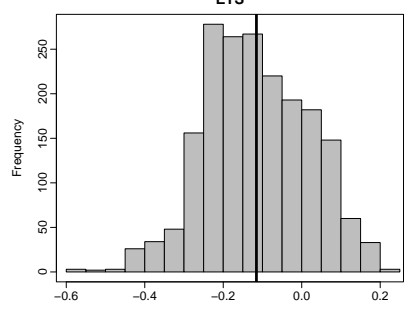 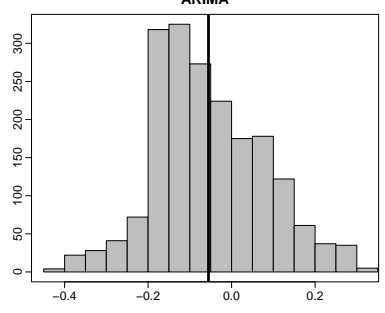 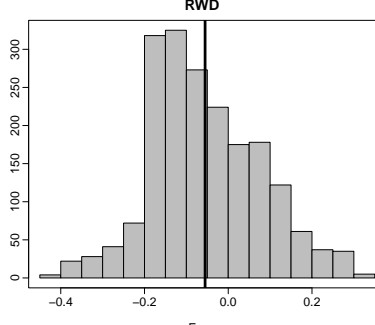

**Figure 5.** *Cont.*

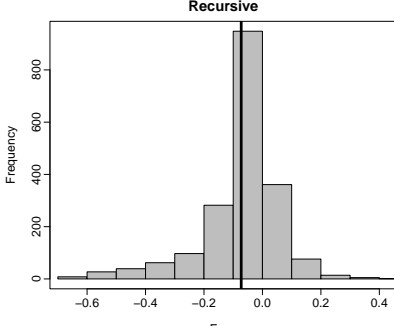
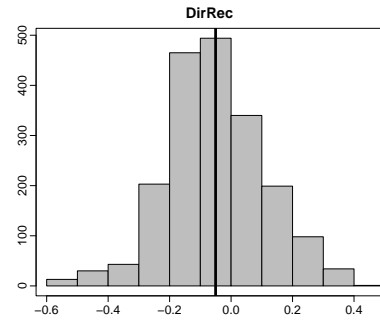

**Figure 5.** Histograms of pointwise 20-step-ahead forecast errors (black vertical lines denote the mean of the pointwise forecast errors) were obtained by all the methods for the log mortality dataset of Australia (female).

The difference between the forecast accuracy of the methods in the results in Figures 2–5 is mainly based on the forecasts of the principal component scores. A graphical display of the forecasted principal component scores obtained by all the methods from the log mortality datasets is presented in Figure 6. This figure shows that the overall trend of the principal component scores is better captured by the forecasts obtained from the machine learning-based methods than those of the classical time series methods. This result, in turn, causes the proposed methods to produce improved forecasts than the traditional FTS methods. In more detail, the forecasted principal component scores in Figure 6 can be seen as an indication of the point forecast accuracy results of the methods presented in Figures 2 and 3. For example, from Figure 6, the forecasted principal component scores of the ETS-based FTS method are far from those of other methods (and far from the general trends of the time series of principal component scores) for datasets from Australia and the UK. These results lead to worse forecasting accuracy for the ETS-based FTS method, as can be seen from Figure 6. On the other hand, the ETS-based method produces relatively closer forecasts for the principal component scores to those of the other two traditional FTS methods for the datasets from Canada. The reflection of this result can be seen in Figures 2 and 3 so that all the traditional FTS methods produce similar forecast accuracy results.

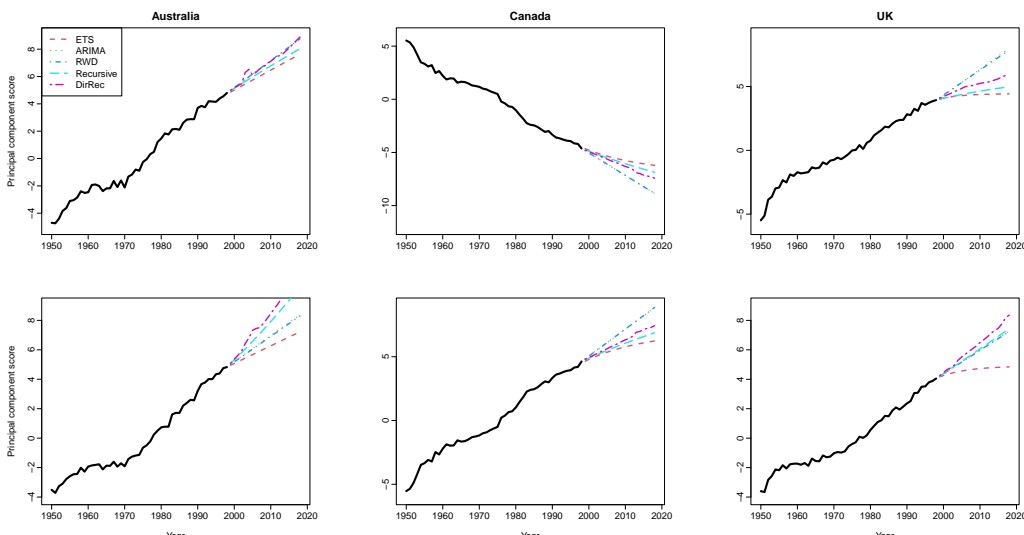

**Figure 6.** Computed principal component scores (solid black lines) for the first functional principal component and their 20-step-ahead forecasts (colored lines). The principal component scores and their forecasts are obtained from the log mortality datasets (female-first row and male-second row) of Australia (**left panel**), Canada (**middle panel**), and UK (**right panel**).

The results for the interval forecast accuracy of the methods are presented in Figure 7. All the methods produce similar interval score values for short and mid-term forecast horizons from this figure. In contrast, the proposed methods generally produce smaller interval score values than those obtained by the classical FTS methods for long-term forecast horizons. The interval forecast accuracy performance of the methods is similar to their point forecast accuracy performance as presented in Figures 2 and 3. The superior long-term interval forecast accuracy of the proposed methods over the traditional FTS methods is more apparent for Australia and the UK. In contrast, all the methods generally produce similar interval forecast accuracy for the Canadian age-specific mortality rate dataset. Similar to the discussions presented in the previous paragraph, this result can also be explained by the forecasted principal component scores presented in Figure 6. Because the forecasted principal component scores obtained from the proposed methods (especially from the recursive-based FTS method) better capture the overall trend of the time series of the principal component scores, they produce improved interval score accuracy than those of other methods.

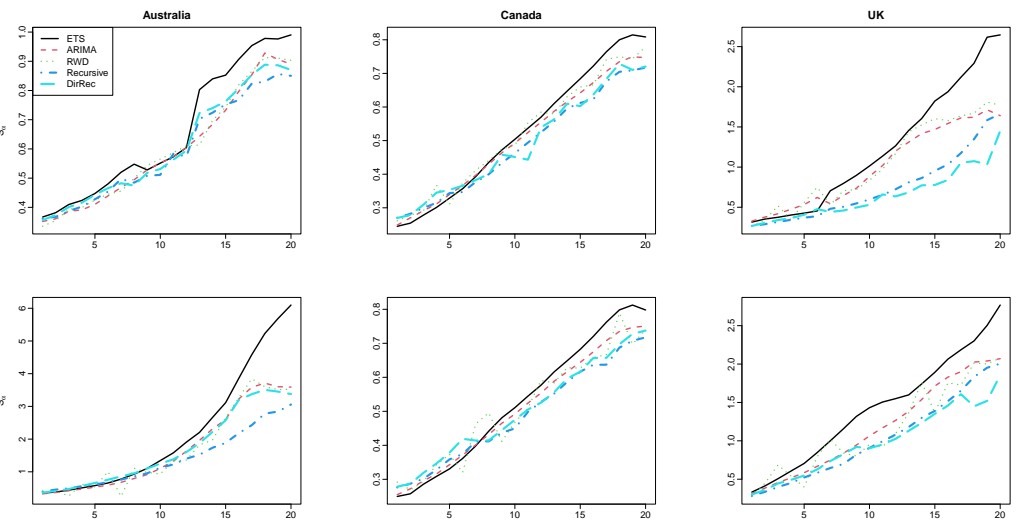

**Figure 7.** Computed $S_\alpha$ values of log mortality rates for Australia (**first column**), Canada (**second column**), and the UK (**third column**). The results for the female group are given in the first row, while the results for the male group are given in the second row.

## 6. Conclusions

Mortality modeling and accurate forecasting of age-specific mortality rates are important problems for governments and insurance industries to manage long-term plans, such as sustainability of pensions and determining fixed-term or life annuity prices (see e.g., Shang and Haberman, 2020 [39]). FTS method is one of the more attractive methods used to obtain mortality forecasts. As an extension of LC-based methods, several FTS methods have been proposed. However, all the existing FTS methods are based on the traditional time series models, which are data-dependent, so different estimation strategies and univariate time-series forecasting models may be required for different datasets. In addition, most of the existing FTS methods are suitable for short-term forecasts. However, multi-step-ahead mortality forecasts may be more useful for policymakers to make long-term plans.

This paper proposes an FTS method based on machine-learning-based multi-step-ahead forecasting strategies. The smooth mortality curves are decomposed using the DFPCA method in the proposed methods. Automatic learning algorithms are used to obtain the future realizations of the extracted dynamic principal component scores. One-to-twenty-step-ahead point and interval forecast accuracy of the proposed methods are evaluated using six datasets from three countries (Australia, Canada, and the UK) compared with four existing FTS methods. Our results demonstrate that the proposed methods produce similar point and interval forecast results to traditional FTS methods for short and moderate-term

forecast horizons. On the other hand, they produce improved point and interval forecast accuracies compared with existing FTS methods for long-term forecast results.

The proposed methodology can be extended further, and here, we briefly mention three ways: (1) In this study, we only present the usefulness of the proposed method for forecasting age-specific mortality rates. The proposed method can be used for forecasting subnational age-specific mortality rates as an alternative to the methods proposed by Shang and 270 Haberman (2020) [39] and Shang and Yang (2021) [25]; (2) the proposed methods can also be applied to other types of mortality, such as cause-specific mortality rates; (3) finally, the performance of the proposed methods can be further improved by using other multi-step-ahead forecasting strategies, such as multi-input–multioutput and its combination with direct strategy (see, e.g., Taieb et al., 2012 [27] for details).

**Author Contributions:** Conceptualization, U.B. and H.S.; methodology, U.B. and H.S.; software, U.B. and H.S.; validation, U.B. and H.S.; formal analysis, U.B. and H.S.; investigation, U.B. and H.S.; resources, U.B. and H.S.; data curation, U.B. and H.S.; writing—original draft preparation, U.B. and H.S.; writing—review and editing, U.B. and H.S.; visualization, U.B. and H.S.; supervision, H.S.; project administration, U.B. and H.S. All authors have read and agreed to the published version of the manuscript.

**Funding:** This research was funded by The Scientific and Technological Research Council of Turkey (TUBITAK) grant number 120F270.

**Institutional Review Board Statement:** Not applicable.

**Informed Consent Statement:** Not applicable.

**Data Availability Statement:** All the datasets used in this study are available at https://www.mortality.org/ (accessed on 31 January 2022).

**Conflicts of Interest:** The authors declare no conflict of interest.

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
