# Peer review of "Machine-Learning-Based Functional Time Series Forecasting: Application to Age-Specific Mortality Rates"

_forecasting, doi:10.3390/forecast4010022_

Round 1

Reviewer 1 Report

The authors propose a functional time series method to obtain accurate multi-step-ahead forecasts for age-specific mortality rates. They employ the dynamic functional principal component analysis method to decompose the mortality curves and use some Machine learning-based multi-step-ahead forecasting strategies to learn the underlying structure of the data automatically. The numerical results confirm the effectiveness of the proposed approach.  

The manuscript is well-written, and the methodological part is present in a formal but clear way. Some points of the experiments part should be improved: 

  1. Australia, Canada, and UK mortality are investigated. Which criterion has been used to select the countries? If possible, motivate this choice.
  2. The differences in the error curves in Figure 3 should be further discussed. In addition, the evidence shown in Figures 5 and 6 could be described in more detail, for example, discussing the differences among countries. 
  3. Please also consider the Mean Absolute Error to evaluate the point forecasts of the different methods. This could highlight other differences among the methods.

Minor comments: 

  1. The caption of figure 1 is wrong.
  2. Some authors have already considered machine learning and deep learning in mortality forecasts. Some references to these approaches could be included in the literature review.

Author Response

Please see the attached response letter. We have addressed your comments under Reviewer 1.

Reviewer 2 Report

The accurate forecasting of age-specific mortality rates is an important problem for many developed countries’ governments and insurance industries to do long-term plans. Most of the existing FTS methods are based on the traditional time series models, and are data-dependent. They are suitable for short-term forecasts bot not long-term forecasts. This paper proposes an FTS method based on machine learning-based multi-step-ahead forecasting strategies. The DFPCA method and automatic learning algorithms are imported into the proposed method. Six datasets from Australia, Canada, and the UK are inputted. Their experimental results show that the proposed methods improved the point and interval forecast accuracies compared with existing FTS methods for long-term forecast results. The methods and case studies presented in this paper show that it is a meaningful solution on forecasting of age-specific mortality rates.

The last group dataset is at and beyond 95. You said it will avoid erratic data. Why the number is 95, but not another number? What is your reason for choosing it? Please explain in detailed.

Please add the pseudo-codes for the proposed method in the paper.

Please add the normal scale results for figure 1. They are about Denmark, Netherlands and Switzerland ???  They are not in your data sets. Are they really your experimental results?

Please show the normal scale figures for the ages from 0 to 95. And which years’ data are selected in your comparing test?

Author Response

Please see the attached response letter, we have addressed your comments under Reviewer 2.
